# Alternative polyadenylation mediates genetic regulation of gene expression

**Briana E Mittleman[1], Sebastian Pott[2], Shane Warland[3], Tony Zeng[3], Zepeng Mu[1], Mayher Kaur[3], Yoav Gilad[2,3], Yang Li[2,3]***

[1]Genetics, Genomics, and Systems Biology, University of Chicago, Chicago, United States; [2]Department of Human Genetics, University of Chicago, Chicago, United States; [3]Section of Genetic Medicine, Department of Medicine, University of Chicago, Chicago, United States

**Abstract** Little is known about co-transcriptional or post-transcriptional regulatory mechanisms linking noncoding variation to variation in organismal traits. To begin addressing this gap, we used 3' Seq to study the impact of genetic variation on alternative polyadenylation (APA) in the nuclear and total mRNA fractions of 52 HapMap Yoruba human lymphoblastoid cell lines. We mapped 602 APA quantitative trait loci (apaQTLs) at 10% FDR, of which 152 were nuclear specific. Effect sizes at intronic apaQTLs are negatively correlated with eQTL effect sizes. These observations suggest genetic variants can decrease mRNA expression levels by increasing usage of intronic PAS. We also identified 24 apaQTLs associated with protein levels, but not mRNA expression. Finally, we found that 19% of apaQTLs can be associated with disease. Thus, our work demonstrates that APA links genetic variation to variation in gene expression, protein expression, and disease risk, and reveals uncharted modes of genetic regulation.

## Introduction

Most genetic variants associated with complex traits are noncoding, suggesting that inter-individual variation in gene regulation plays a dominant role in determining phenotypic outcome. To investigate the function of trait-associated variants identified using genome-wide association studies (GWAS), studies have used regulatory quantitative trait loci (QTL) mapping to associate GWAS loci with variation in mRNA expression levels, DNA methylation levels, and other molecular phenotypes. Although many GWAS loci affect mRNA expression levels (i.e. are eQTLs), several recent discoveries highlight the pressing need for a better understanding of the genetic control of gene regulation, beyond that of just mRNA expression levels. For example, one recent study (*Chun et al., 2017*) found that the majority of autoimmune GWAS loci do not appear to affect mRNA expression levels. Two other studies observed that many genetic variants that affect protein expression levels (pQTLs) do not affect mRNA expression levels (*Battle et al., 2015*; *Chick et al., 2016*). Specifically, Battle and colleagues found that about half of the cis-pQTLs they identified in human LCLs (146 out of 278, 52%) did not appear to impact gene expression levels in the same lymphoblastoid cell lines (LCLs) (*Battle et al., 2015*). Altogether these findings indicate that there may be unknown or understudied regulatory mechanisms that link genetic variation to complex traits, and that these mechanisms are independent of changes in the amplitude of mRNA expression levels. Moreover, even when a disease-associated variant impacts mRNA expression levels, the mechanisms by which expression is affected are often unclear. Indeed, a third of all eQTLs identified in human LCLs are not associated with variation in chromatin as measured using assays for chromatin accessibility or for modification levels of several histone marks (*Li et al., 2016*). These observations raise the possibility that understudied regulatory mechanisms mediate the effect of a substantial number of genetic variants on gene expression level.

**\*For correspondence:**
yangili1@uchicago.edu

**Competing interests:** The authors declare that no competing interests exist.

One such understudied mechanism is alternative polyadenylation (APA). Well over half of all human protein coding genes encode multiple polyadenylation sites (PAS), resulting in the production of diverse mRNAs with alternative termination sites (*Tian and Manley, 2017*; *Mayr, 2016*; *Shi, 2012*). Unlike alternative mRNA splicing, which leads to changes in splice site selection, APA leads to changes in the transcript termination site, often resulting in 3′ untranslated regions (UTRs) with different lengths. As 3′ UTRs are densely packed with regulatory elements that impact mRNA stability, miRNA binding, and mRNA localization (reviewed in *Mayr, 2017*; *Tian and Manley, 2017*), genetic control of APA may be a key mechanism by which genetic variants impact gene regulation, including mRNA expression levels, without affecting chromatin-level phenotypes such as promoter or enhancer activity. Moreover, proteins translated from different APA isoforms may differ in length and protein-protein interactions, and these differences can impact cellular phenotype. For example, globally increased usage of intronic PAS has been shown to increase risk for multiple myeloma and chronic lymphocytic leukemia through the translation of truncated mRNAs into truncated proteins, which impairs tumour-suppressive functions (*Lee et al., 2018*; *Singh et al., 2018*).

To evaluate the role of APA in mediating genetic effects on gene expression and disease, we sought to identify genetic variants associated with APA on a genome-wide scale. To date, the few studies that have used genome-wide methods to identify variants associated with APA (apaQTLs) have used existing RNA-seq data to infer PAS locations and usage (*Li et al., 2019*; *Yoon et al., 2012*; *Yang et al., 2020*; *Bonder et al., 2019*; *Mariella et al., 2019*). While using existing RNA-seq to study APA is economical, identifying PAS and estimating usage using RNA-seq are error-prone and often imprecise (*Ha et al., 2018*). Furthermore, using standard RNA-seq data alone to study APA is not informative with regard to whether inter-individual differences in PAS usages are the result of variation in transcriptional termination site choice, or isoform-specific decay or export. Here, we used 3′ RNA-seq (3′ Seq) to measure PAS usage in steady-state mRNA collected from whole cells as well as mRNA collected from the nucleus, which is comprised of a high proportion of nascent mRNA. This design allowed us to study the effect of genetic variation on isoform PAS at multiple stages of the mRNA lifecycle. Importantly, we collected these data from a panel of human lymphoblastoid cell lines (LCLs) that were previously profiled in great molecular detail, including measurements at the chromatin, RNA, and protein levels (*Degner et al., 2012*; *McVicker et al., 2013*; *Li et al., 2016*; *Pickrell et al., 2010*). Integrating the apaQTLs we identified with previously collected molecular data allowed us to study the impact of APA variation on the major steps of the gene regulatory cascade (*Figure 1A*). We use these data to show that genetic effects on APA can affect virtually all steps of gene regulation (mRNA expression level, translation rate, and protein expression level), and that such effects can impact protein expression, without affecting RNA expression.

## Results

### Alternative polyadenylation in human LCLs as defined using nuclear and total mRNA 3′ Seq

To measure inter-individual variation in APA, we quantified PAS usage in a panel of 52 Yoruba Hap-Map LCLs. These same cell lines have been the subjects of multiple studies of gene regulation over the last decade (*Degner et al., 2012*; *McVicker et al., 2013*; *Li et al., 2016*; *Pickrell et al., 2010*). We applied 3′ Seq to mRNA collected from whole cells (total mRNA fraction) of 52 LCLs and used a peak calling approach (Materials and methods) to comprehensively identify PAS and estimate their usage (*Figure 1B–C*). Our approach obviates the need for existing annotations, which are biased towards highly expressed isoforms or isoforms expressed in well studied cell-types with higher RNA-seq coverage. In addition, to capture polyadenylated mRNA that may be under-represented or absent in the total mRNA fraction due to rapid turnover, we separately applied 3′ Seq to mRNA from isolated nuclei (nuclear fraction) of the same 52 LCLs. We found that the ratio of the number of nuclear 3′ Seq read count to total mRNA 3′ Seq read count is positively correlated with two different measures of mRNA decay (*Supplementary file 1*), indeed supporting our expectation that 3′ Seq applied to the nuclear mRNA fraction captured RNA transcripts at an earlier stage of RNA processing. Because 3′ Seq uses polyA priming to capture the location of polyadenylation sites and is therefore prone to internal priming at transcribed regions that are A-rich, we carefully filtered our data to

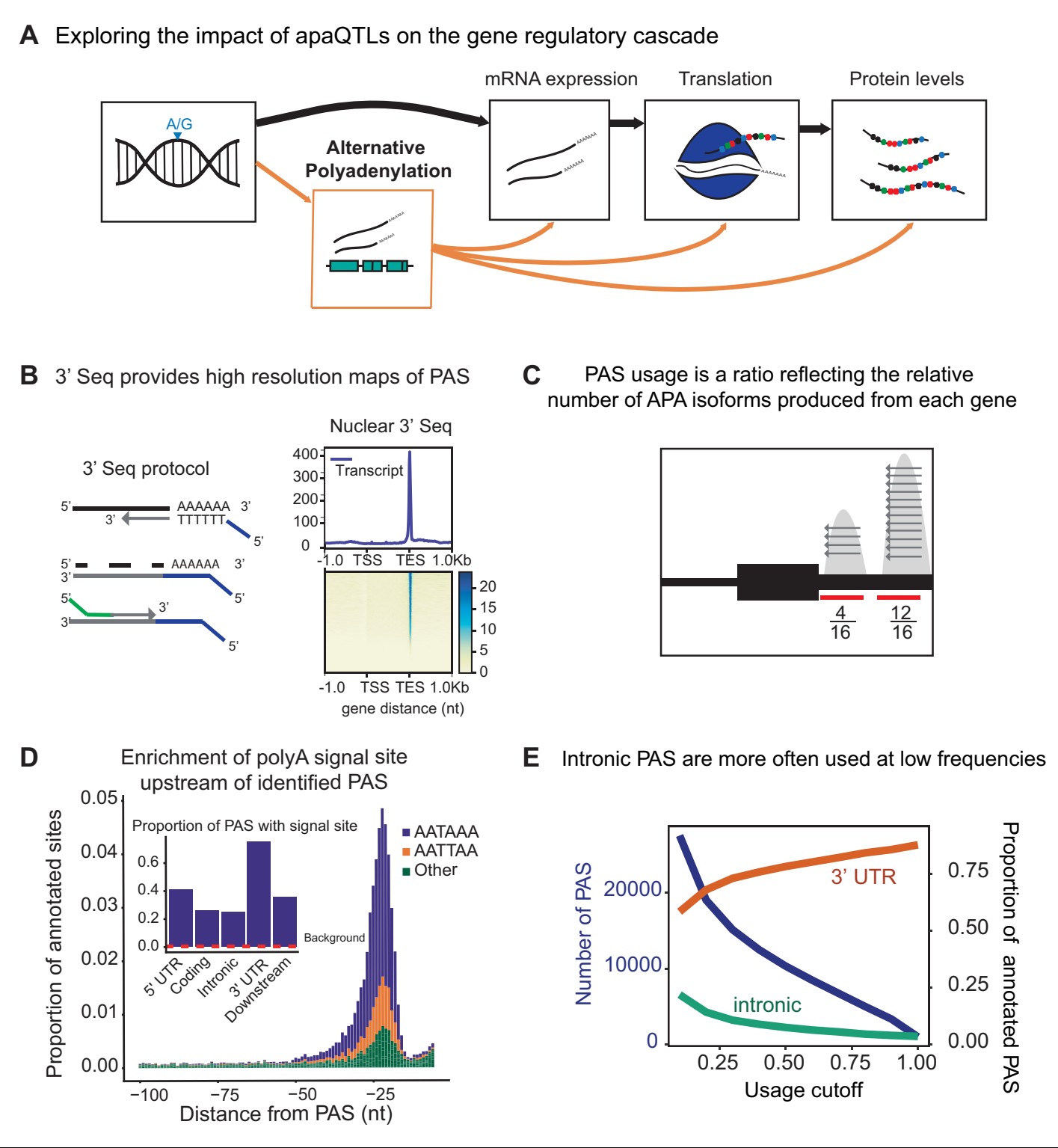

**Figure 1.** 3′ sequencing of mRNA from the nuclear fraction to study inter-individual variation in APA. (**A**) Schematic of how genetic variants affect phenotypes by percolating through gene regulatory layers (black arrows). We aimed to understand how genetic variation can mediate gene regulation through alternative polyadenylation (orange arrows). (**B**) (*Left*) Schematic of Lexogen QuantSeq Reverse 3′ mRNA-Seq protocol (*Moll et al., 2014*) (*Right*) Meta gene plot showing read coverage for five 3′ Seq libraries collected from nuclei isolated from LCLs. (**C**) Representation for how PAS usage is calculated. Read count for each PAS were divided by the total number of reads at all PAS for the gene. (**D**) (*Main*) Stacked density of canonical (AATAAA, AATTAA) and other polyadenylation signal sites (AAAAAA, AAAAAG, AATACA, AATAGA, AATATA, ACTAAA, AGTAAA, CATAAA, *Figure 1 continued on next page*

*Figure 1 continued*

GATAAA, TATAAA) upstream of identified PAS. (*Inset*) Proportion of PAS in different genomic regions with a polyadenylation signal site 10–50 bp upstream of cleavage site. The red dotted line represents the proportion of signal site in random 40 bp windows, i.e. the intronic background. (**D**) The proportion of intronic PAS increases as the usage cutoff decreases, implying that a disproportionate number of intronic PAS are used at low frequencies. The blue line represents the number of PAS identified as the stringency of the usage cutoff increases. The orange and green lines represent the proportion of PAS in the 3′ UTR and introns, respectively.

The online version of this article includes the following figure supplement(s) for figure 1:

**Figure supplement 1.** Relationship between the number of PAS identified in our study and gene expression levels (TPM) as measured from GEUVADIS YRI LCLs.

**Figure supplement 2.** Intronic and 3′ UTR share signal site motif density distributions.

**Figure supplement 3.** Number of PAS identified with usage larger than the usage cutoff (x-axis) in the total mRNA fraction (purple).

**Figure supplement 4.** Intronic PAS are enriched in introns with the weakest 5′ splice sites.

**Figure supplement 5.** We adapted *LeafCutter* to identify genes with significant differential usage of PAS between the total and nuclear fraction (*Li et al., 2018*).

**Figure supplement 6.** Our identified PAS include both previously annotated and novel sites.

**Figure supplement 7.** Validation of cellular fractionation with western blots.

**Figure supplement 8.** Proportion of reads that map to the genome (mapped) and the proportion of final reads used for analysis are cleanly mapped (Clean Mapped) by nuclear mRNA library.

**Figure supplement 9.** Proportion of reads that map to the genome (mapped) and the proportion of final reads used for analysis that are cleanly mapped (Clean Mapped) by total mRNA library.

**Figure supplement 10.** Total number of reads that map to the genome (mapped) and the number of final reads used for analysis that are cleanly mapped (Clean Mapped) by nuclear mRNA library.

**Figure supplement 11.** Total number of reads that map to the genome (mapped) and the number of final reads used for analysis that are cleanly mapped (Clean Mapped) by total mRNA library.

ensure a minimal effect of mispriming on the set of PAS we considered (Materials and methods). Specifically, similar to methods previously described, we filtered both individual reads and PAS that map to genomic regions with 70% A nucleotides or a stretch of 6 A's in the 10 nucleotides upstream (*Sheppard et al., 2013*; *Tian et al., 2005*). After quality control and filtering, we defined the usage of each PAS in a sample as the ratio of the number of reads that map to the PAS to the number of reads that map to all PAS for the same gene (*Figure 1C*) (Materials and methods). Thus, we measure the usage of a PAS as the fraction of transcripts using that PAS over the total number of transcripts from the same gene.

We identified 41,810 nuclear PAS in 15,043 genes with at least 5% mean usage across the 52 LCL samples. We found that 67% of the protein coding genes expressed in LCLs harbor multiple PAS, suggesting that APA can impact the regulation of most genes (*Tian and Manley, 2017*; *Mayr, 2016*; *Shi, 2012*). Interestingly, we identified a slight negative correlation between the expression level of a gene and the number of PAS identified for the gene (Pearsons Correlation = −0.12, $p = 2.2 \times 10^{-16}$). In particular, genes with a single PAS tend to be expressed more highly than genes with multiple PAS. This observation is counter-intuitive from a statistical perspective, and it shows that, in general, our ability to detect PAS was not limited by 3′ Seq coverage (*Figure 1—figure supplement 1*, Materials and methods). We found that the polyA binding protein motif (AATAAA), also known as the polyadenylation signal site, is the most strongly enriched motif in the 50 bp regions upstream of our PAS (hypergeometric test, $p < 10^{-391}$).

We observed that PAS in the 3′ UTR are more likely to have a polyadenylation signal compared with intronic PAS ($p < 10^{-16}$, difference of proportion t-test, 75.0% vs 24.8%,) (*Figure 1D*, *Figure 1—figure supplement 2*) and that nearly half (48.3%) of all 41,810 PAS we identified are located in 3′ UTRs (19.4x enrichment) (*Singh et al., 2018*). Nevertheless, despite an overall depletion of PAS in introns (0.35x genome-wide levels), we found that the number of PAS in introns is notable (12,793/41,810; 30.6%) (*Figure 1E*, *Figure 1—figure supplement 3*). While signal sites were more highly enriched near 3′ UTR PAS than intronic PAS, PAS in introns show clear enrichment of polyadenylation motif 10–50 bp upstream of the cleavage site compared to background intronic sequences (24.8% vs 0.24% $p < 10^{-16}$, difference of proportion t-test, *Figure 1D*). Thus, the recognition of intronic polyadenylation signals is a general mechanism that can result in premature termination of transcription.

We tested the hypothesis that the intronic PAS we identified correspond to truncated mRNA transcripts that escaped telescripting, whereby the U1 snRNP protects introns from premature cleavage and polyadenylation (*Kaida et al., 2010*; *Berg et al., 2012*; *Oh et al., 2017*). Because the main role of U1 snRNP is to bind and recognize 5' splice sites, the telescripting model predicts that weaker 5' splice sites can result in decreased U1 snRNP affinity for an intron and thus higher rates of early cleavage and polyadenylation. We estimated 5' splice site strength for all introns using MaxEntScore (*Yeo and Burge, 2004*) and found that introns with the weakest 5' splice sites harbored more PAS than introns with stronger 5' splice sites (1.5x fold difference, first decile vs remaining deciles, hypergeometric test $p = 8.07 \times 10^{-87}$, *Figure 1—figure supplement 4*, Materials and methods) (*Tian et al., 2007*). Moreover, we found that the top 10% of most highly used intronic PAS have weaker 5' splice sites than introns with lowly used PAS or a random set of introns (Mean MaxEntScore 6.43 vs 7.26 vs 7.48, $p = 1.4 \times 10^{-3}$, Wilcoxon rank sum test). These observations are consistent with the hypothesis that telescripting protects nascent transcripts from early cleavage and polyadenylation in introns, and that the intronic PAS we observe result from transcripts that escape telescripting (*Kaida et al., 2010*; *Berg et al., 2012*; *Oh et al., 2017*).

We observed that intronic PAS have on average lower usage across individuals than PAS located in 3' UTRs (16.9% vs 46.2%). Lower usage of intronic PAS may be explained by weaker polyadenylation signals at intronic PAS compared to 3' UTR PAS or by the impact of telescripting on intronic polyadenylation. However, we hypothesized that some intronic PAS have low usage because premature polyadenylation at intronic sites can produce short-lived transcripts that are rapidly degraded and thus are under-represented in the total mRNA fraction. To test this hypothesis, we identified PAS that are used more often, or exclusively, in the nuclear fraction compared to the total mRNA fraction. By comparing PAS usage estimated in the nuclear and total mRNA fractions from all 52 individuals, we identified at 10% FDR 591 PAS in 585 genes that are used at least 20% more in the nuclear compared to the total mRNA fraction. Of these 591 PAS, 134 were found to be used by 1% or less of the transcripts in the total mRNA fraction, suggesting that these transcripts may be absent from the cytoplasm (*Figure 1E*, *Figure 1—figure supplement 5*, Materials and methods). Notably, we found that 387 of the nuclear-enriched PAS are intronic (*Figure 1—figure supplement 5*), a large proportion of which (83.4% vs 43% for all PAS) are absent from a comprehensive annotation of PAS compiled from 78 human studies that used 3' Seq (Materials and methods, *Figure 1—figure supplement 6*; *Wang et al., 2018*). While no other study has directly measured PAS usage in nuclei, a proportion of the nuclear enriched intronic sites have been identified in a number of human tissues (up to 10%, *Supplementary file 1*). These findings suggest that mRNA transcripts are terminated and polyadenylated in introns at a higher frequency than generally appreciated, and that many of these isoforms escape detection from studies of total mRNA fraction owing to their rapid decay or their propensity to remain within the nucleus.

## Genetic loci associated with variation in APA

Having established that APA can contribute to the generation of complex transcript isoforms, we sought to identify genetic loci associated with inter-individual variation in APA. We normalized each PAS usage ratio using LeafCutter (*Li et al., 2018*) and tested *cis*-associations between genetic variants and PAS usage, correcting for batch and the top principal components (Materials and methods, *Figure 2—figure supplements 1–3*). Using 3' Seq data from the nuclear fraction, we identified 602 nuclear apaQTLs in 479 genes at 10% FDR. In the total mRNA fraction, we identified 443 apaQTLs in 353 genes at 10% FDR. For example, individuals with the C/C genotype (rs11032578) show higher usage of an intronic PAS in the *ABTB2* gene compared to individuals that are heterozygous C/T or homozygous T/T (*Figure 2A*). In both fractions, apaQTL lead SNPs are enriched near the PAS they most strongly correlate with and near the 3' ends of gene bodies (*Figure 2B*, *Figure 2—figure supplement 4*). The proximity of the apaQTL lead SNPs to PAS may suggest that genetic variants that affect polyadenylation signal motifs drive most of the genetic effects on APA. Although we observed an enrichment of apaQTLs in signal motifs, genetic variants that alter signal motifs are unlikely to explain the majority of apaQTLs (*Figure 2—figure supplement 5*).

Our study design provides the unique opportunity to evaluate the likely mechanisms by which genetic variation controls PAS usage. While previous studies have demonstrated that genetic

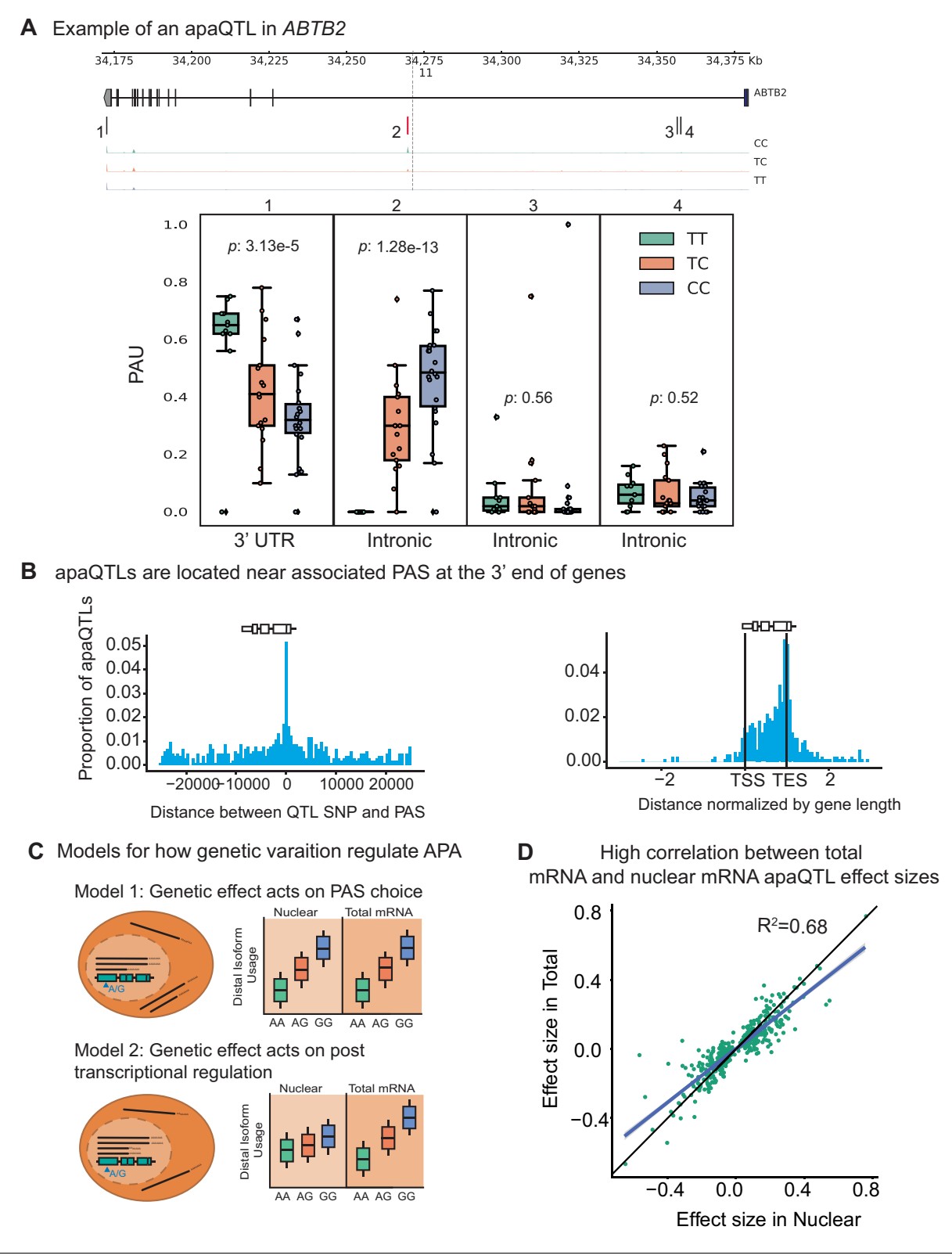

**Figure 2.** Impact of genetic variation on PAS choice. (**A**) An apaQTL in the *ABTB2* gene impact usage of an intronic PAS. (*Top*) Gene track and identified PAS. Each bar represents a PAS. The red bar corresponds to the PAS most strongly associated with the apaQTL. The vertical dotted line represents the position of the lead apaQTL SNP. (*Bottom*) Boxplot of polyadenylation site usage at each PAS by genotype listed according to the PAS order above. The C allele increases usage of the intronic PAS. (**B**) (*Left*) Location of the lead nuclear apaQTL SNPs relative to their corresponding PAS. *Figure 2 continued on next page*

*Figure 2 continued*

(*Right*) Meta gene plot showing the distribution of apaQTL SNPs in the annotated gene body, where 0 represents the transcription start site and 1 represents the annotated transcription end site. (C) Two mechanistic models for how genetic variants can affect PAS usage. (*Model 1*) Genetic variation acts directly on PAS choice. In this case, the apaQTL will be identified with similar effect sizes in both nuclear and total mRNA fractions, or smaller effect size in the total mRNA fraction. (*Model 2*) Genetic variation acts through a post transcriptional mechanism. For example, one mRNA isoform is subject to decay. In this case, the apaQTL will be identified only in the total mRNA fraction, or will be identified in the total mRNA fraction with a larger effect size than in the nuclear mRNA fraction. (D) Effect sizes of apaQTLs originally identified at 10% FDR in the nuclear mRNA fraction plotted against the effect sizes ascertained in the total mRNA fraction. Regression line is shown in blue and $y = x$ line is shown in black.

The online version of this article includes the following figure supplement(s) for figure 2:

**Figure supplement 1.** Q-Q plots for total and nuclear apaQTL linear regression tests.

**Figure supplement 2.** Proportion of PAS in different genomic locations with a significant apaQTL.

**Figure supplement 3.** Top 4 PCs included in our apaQTL linear models to account for technical variation.

**Figure supplement 4.** Expansion of *Figure 4B* that includes both fractions.

**Figure supplement 5.** Signal site disruption cannot explain the majority of apaQTLs.

**Figure supplement 6.** Total specific apaQTLs have smaller effect sizes than shared apaQTLs.

**Figure supplement 7.** Storey's Pi statistics suggest most apaQTLs are shared between fractions.

**Figure supplement 8.** Sharing of genetic effects on APA between fractions.

variants can impact PAS usage, it has been difficult to discern whether the variation in PAS usage is primarily driven by genetic effects on cleavage and polyadenylation (*Figure 2C*, Model 1), or on the mRNA lifecycle (e.g. by impacting miRNA binding sites and decay) (*Figure 2C*, Model 2). We reasoned that if genetic effects functioned primarily by affecting post-transcriptional regulation such as decay or export, then this effect would be detectable in the total mRNA fraction, but would be smaller or undetectable in the nuclear mRNA fraction (*Supplementary file 1*). Interestingly, we found that only 97 apaQTLs (of 443 apaQTLs, 21.9%) identified in the total mRNA fraction were not detected in the nuclear mRNA fraction and these associations are much weaker than shared apaQTLs (*Figure 2—figure supplement 6*). We thus suspect that we currently lack statistical power to detect most of these 97 apaQTLs in the nuclear mRNA fraction. To estimate sharing of apaQTLs across the two mRNA fractions, we used Storey's $\pi_0$ statistics and found that the vast majority of apaQTLs identified in the total mRNA fractions were estimated to also affect PAS usage in the nuclear mRNA fraction ($\pi_1$=0.87, *Figure 2—figure supplement 7*). In addition, we found that the genetic effect sizes on PAS usage were very similar across the two mRNA fractions ($r^2$=0.66; $p < 10^{-16}$, *Figure 2D*, *Figure 2—figure supplement 8*). Altogether these observations show that most genetic variants impact PAS usage by affecting polyadenylation site choice. Supporting this notion, we found weak or no enrichment of apaQTLs in sites bound by RNA binding proteins as identified using eCLIP data from ENCODE (*Supplementary file 1*).

## Impact of apaQTLs on gene expression levels

While we believe that nearly all genetic variants impact PAS usage by affecting polyadenylation site choice and not isoform-specific decay or export, this model is not incompatible with a model in which genetic variants can sometimes impact expression by affecting APA. For example, a genetic variant might increase the relative production of an isoform that is less stable, in which case total transcript levels would decrease. Therefore, next, we asked whether genetic variants could impact gene expression levels by direct effects on APA. We hypothesized that this mode of genetic regulation may be prevalent, in particular for genes with intronic PAS, because isoforms using intronic PAS are often subject to rapid decay. In this model, the genetic effect changes the relative production of isoforms with different relative stabilities rather than specifically modulating the stability of an isoform, for example by increasing affinity for microRNA binding in the 3' UTR.

To test this hypothesis, we focused on the set of 602 apaQTLs that we identified in the nuclear mRNA fraction, representing genetic variants that impact PAS choice. Our hypothesis predicts that genetic variants that increase intronic PAS usage should decrease gene expression levels. In line with this prediction, we found a negative correlation between the genetic effect sizes for intronic PAS usage and mRNA expression levels ($p = 8.97 \times 10^{-7}$, *Figure 3A*, *Figure 3—figure supplement 1*). Thus, our analysis suggests a widespread mechanism whereby genetic variants decrease mRNA expression levels by increasing choice of isoforms with premature PAS that are subject to rapid

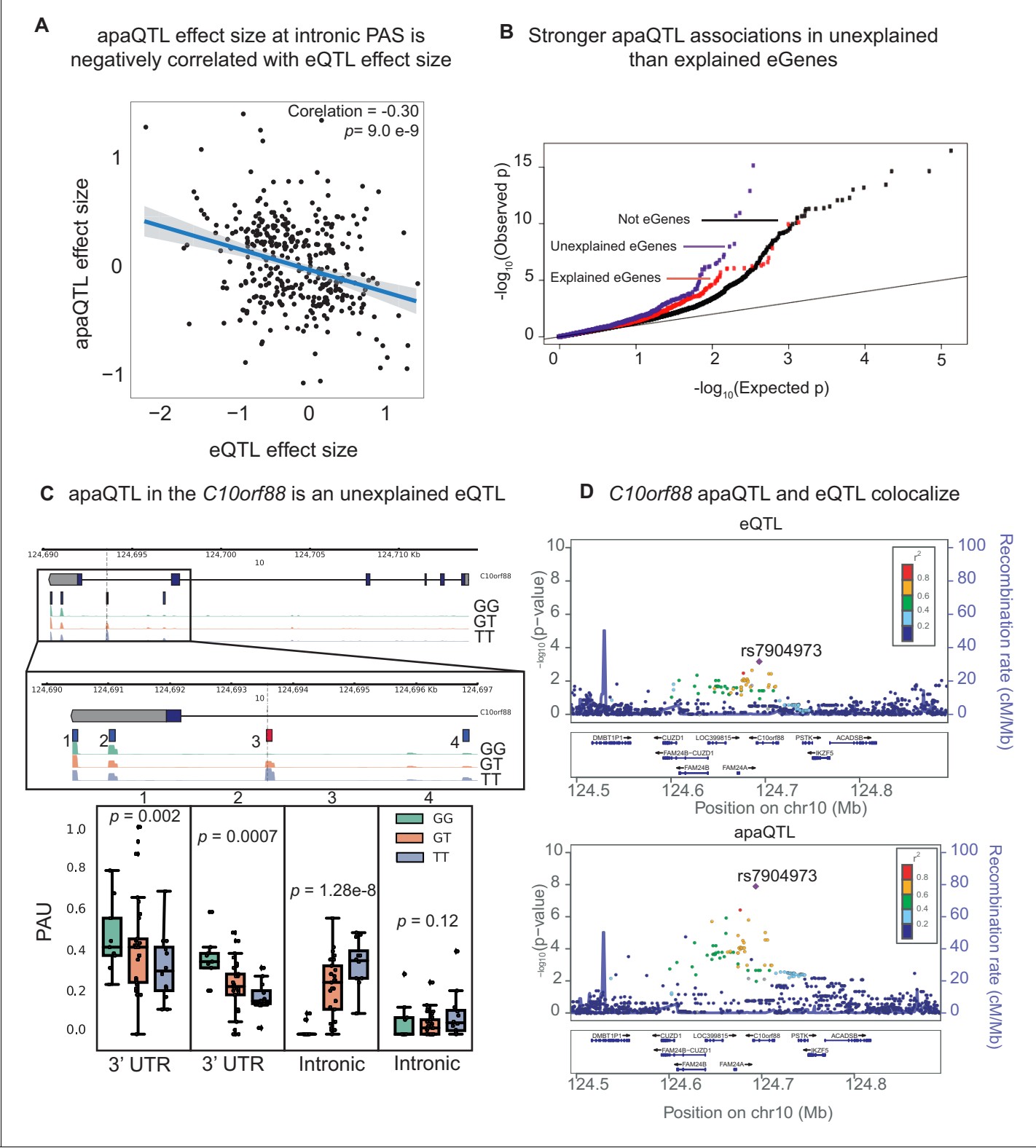

**Figure 3.** APA can mediate genetic effects on mRNA expression. (**A**) Scatter plot of intronic apaQTL effect sizes plotted against their eQTL effect sizes shows negative correlation. (**B**) Quantile-quantile (Q–Q) plot for apaQTLs shows that apaQTLs are more highly enriched in unexplained eGenes (purple dots) compared to explained eGenes (red dots). (**C**) Example of an apaQTL that is also an unexplained eQTL for *C10orf88*. (*Top*) Gene track and identified PAS in the *C10orf88* gene. The red bar corresponds to the PAS most strongly associated with the apaQTL. The vertical dotted line represents the position of the strongest apaQTL SNP. (*Middle*) Zoomed version of track represented above. (*Bottom*) Boxplot of polyadenylation site usage at

*Figure 3 continued on next page*

*Figure 3 continued*

each PAS by genotype listed according to PAS order above. (D) (*Top*) LocusZoom plot for eQTL associations for the *C10orf88* gene. (*Bottom*) LocusZoom plot for apaQTL associations. Interestingly, the lead apaQTL and eQTL SNP, rs7904973, has been linked to increased LDL cholesterol through GWAS (*Klarin et al., 2018*).

The online version of this article includes the following figure supplement(s) for figure 3:

**Figure supplement 1.** Scatter plot showing the relationship between intronic nuclear apaQTL effect size and eQTL effect size after removing outlier SNPs (Filtered for SNPs with eQTL effect size <− 2.0).

**Figure supplement 2.** Q-Q plot showing the total apaQTL (adjusted) p-values separated by whether the gene harbors an explained (red) or unexplained (blue) eQTLs.

**Figure supplement 3.** Bar plot showing the proportion of apaQTLs located in each of the 12 chromatin states from chromHMM.

**Figure supplement 4.** Proportion of eQTLs putatively explained by apaQTLs separated by fraction.

decay. Of interest, we found that 13 apaQTLs that were detected only in the nuclear fraction are also eQTLs, which highlights the importance of considering early stages of the mRNA lifecycle to uncover eQTL mechanisms.

To further investigate the contribution of APA to gene expression, we sought to understand the relationship between apaQTLs and a set of eQTLs that we previously classified as those with explained putative mechanisms, explained eQTLs (1164 eQTLs, ~60%) or as unexplained eQTLs (801 eQTLs, ~40%) using data from the same LCLs (*Li et al., 2016*). The eQTLs with explained putative mechanisms were associated with chromatin-level phenotypes including DNase-I hypersensitivity, histone marks, or DNA methylation, and thus are likely to be mechanistically explained by effects mediated by chromatin-level phenotypes (e.g. enhancer or promoter activity). To test whether apaQTLs might account for unexplained eQTLs, we first asked whether genes with unexplained eQTLs were more likely to also harbor apaQTLs than compared to genes with explained eQTLs. Indeed, we found a significantly higher enrichment of low p-value associations with APA for genes with unexplained eQTLs ($p = 0.01$, *Figure 3B*, *Figure 3—figure supplement 2*) and significantly larger absolute apaQTL effect sizes for unexplained eGenes compared to explained eGenes (0.35 vs. 0.3, Wilcoxon Rank sum test, $p = 6.6 \times 10^{-4}$). We also found that apaQTLs exhibited an association with chromatin states that was more similar to the unexplained eQTLs than the explained eQTLs. In particular, apaQTLs and unexplained eQTLs were more likely to lie in regions of transcription elongation or are associated with weak transcription, and less likely to lie in enhancers or promoters than explained eQTLs (*Figure 3—figure supplement 3*, Materials and methods). Overall, we estimated that 17.3% of otherwise unexplained eQTLs were associated with PAS usage (see Materials and methods). For example, an unexplained eQTL for *C10orf88* (rs7904973) colocalizes with an apaQTL associated with increased usage of an intronic PAS (*Figure 3C*). More generally, we found that eQTLs and apaQTLs colocalize for the majority of genes that had both (Materials and methods, *Supplementary file 1*). This observation thus highlights APA as one important mechanism by which genetic variation impacts gene expression independent from enhancers and promoters.

## APA mediates gene regulation independently of mRNA expression levels

Previous joint analyses of molecular QTLs suggested that functional genetic variants tend to affect gene regulation in a simple and straightforward manner: first impacting chromatin activity, then mRNA expression, and finally protein expression (*Li et al., 2016*; *Battle et al., 2015*). However, because isoforms with different 3' UTRs have been shown to vary in terms of their translation efficiency, we hypothesized that apaQTLs can impact ribosome occupancy and protein expression levels without affecting mRNA expression levels (*Floor and Doudna, 2016*). To test this possibility, we asked whether apaQTLs are enriched among genes without a known eQTL, but that are associated with a ribosome occupancy QTL (riboQTL) or a protein expression QTL (pQTL). Indeed, we found that apaQTLs are enriched among genes with a ribosome QTL (rGenes; Wilcoxon rank sum test $p = 0.01$) and genes with a pQTL (pGenes; $p = 0.0006$) compared to genes with no molecular association (*Figure 4A*, *Figure 4—figure supplement 1*; *Li et al., 2016*; *Battle et al., 2015*). In addition, we observed a small but significant positive correlation between individual variance in APA usage

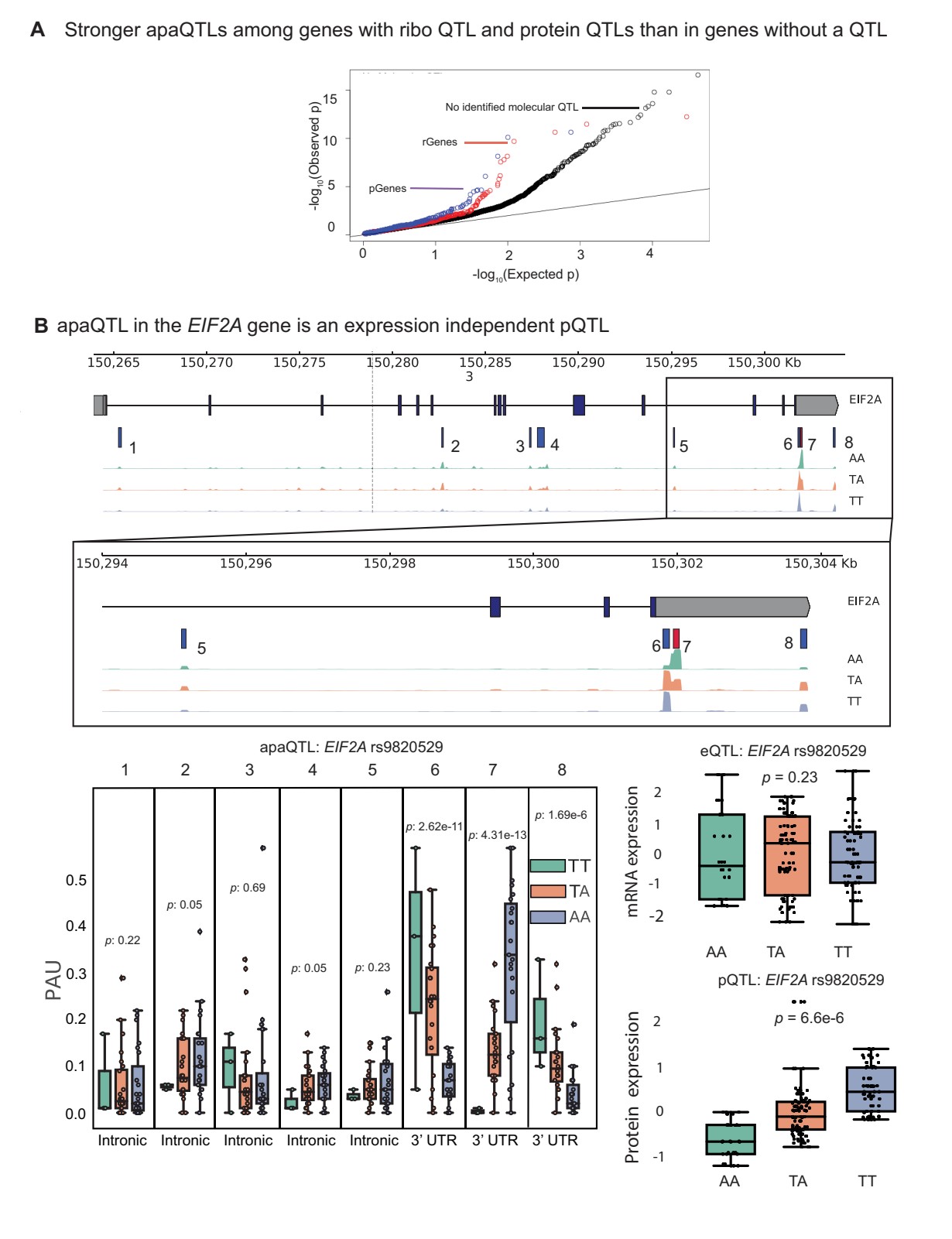

**Figure 4.** apaQTLs can regulate gene expression without affecting mRNA expression levels. (A) Quantile-quantile (Q–Q) plot for apaQTLs separated by genes in previously detected rQTLS (red) and pQTLs (purple) that are not eQTLs. Black points are apaQTL genes with no pQTL, rQTL, or eQTL. (B) (*Top*) Gene track and identified PAS in the *EIF2A* gene. The red bar corresponds to the PAS most strongly associated with the apaQTL. The vertical dotted line represents the position of the strongest apaQTL SNP. (*Middle*) Zoomed version of track represented above. (Bottom Left) Boxplot of

*Figure 4 continued on next page*

*Figure 4 continued*

polyadenylation site usage at each PAS by genotype listed according to PAS order above. (Top Right) Boxplot showing normalized mRNA expression for *EIF2A* by genotype at the apaQTL SNP (rs9820529). (Bottom Right) Boxplot showing normalized protein expression for *EIF2A* by genotype at the apaQTL SNP (rs9820529).

The online version of this article includes the following figure supplement(s) for figure 4:

**Figure supplement 1.** Stronger ribo QTLs and protein QTLs than expression QTLs in total fraction.

**Figure supplement 2.** LocusZoom plots for *EIF2A* apaQTL in *Figure 4B* along with associations with RNA expression, ribosome occupancy (ribo-seq), and protein expression as determined using normalized data from *Li et al., 2016*.

**Figure supplement 3.** Genetic variation around PAS contribute to trait heritability.

and ribosome occupancy (correlation = 0.15, $p < 2.2 \times 10^{-16}$, *Supplementary file 1*), supporting a model in which APA impacts translation efficiency.

In total, we found 24 apaQTLs that affect protein expression, but not mRNA expression (*Supplementary file 2*). Of these, five apaQTLs were significantly associated with ribosome occupancy (*Supplementary file 2*). This finding is particularly noteworthy because nearly all genetic effects on ribosome occupancy have been proposed to be mediated by effects on mRNA expression (*Battle et al., 2015*). Yet, here we provide direct evidence that APA can mediate genetic effects on ribosome occupancy without affecting mRNA expression levels. For example, the apaQTL in the *EIF2A* gene that is associated with a switched usage of two 3' UTR PAS, colocalizes with a pQTL and a ribosome occupancy QTL (*Figure 4B*, *Figure 4—figure supplement 2*), but is not associated with *EIF2A* mRNA levels (*Figure 4B*). Interestingly, the QTL in *EIF2A* affects usage of two PAS in the same 3' UTR implying that the protein sequence encoded by the two isoforms are identical. Thus, the regulatory associations uncovered at *EIF2A* cannot simply be explained by differences in protein isoform stability. Moreover, while differences in 3' UTR are often assumed to play a regulatory function by influencing decay (*Mayr, 2017*), mechanisms involving RNA decay cannot be operational in this case because steady-state mRNA expression is unchanged. Instead, differences between the two isoforms may reflect differential binding of factors that impact translation (*Yamashita and Takeuchi, 2017*), or differential rates of translation re-initiation at the end of a translation cycle (*Rogers et al., 2017*).

We identified 19 pQTLs that were associated with APA but not steady-state gene expression or ribosome occupancy levels. Two previous studies also reported the discovery of pQTLs that were not eQTLs (*Battle et al., 2015*; *Chick et al., 2016*). In both studies, the authors proposed that some genetic effects on protein expression levels were mediated by changes in the protein sequence or by changes in the expression levels of interacting proteins, which would manifest post-translationally. Our finding reveals yet another mode of genetic regulation of protein expression levels by APA (e.g. by affecting recruitment of interacting proteins). Thus, these findings provide clear evidence that APA can affect protein expression levels without affecting gene expression levels. Altogether, our findings suggest complex modes of gene regulation independent of mRNA expression driven by variation in APA.

## APA mediates genetic effects on complex traits

Genetic variation may impact disease risk through APA. We asked whether common variants in the regions around PAS, that is regions enriched for apaQTLs, are enriched in disease heritability. Using LDscore regression to estimate the heritability enrichment of 35 traits in 1 kb regions centered around PAS, we found that 14 of the tested traits were significantly enriched (*Figure 4—figure supplement 3*). Of note, genetic variation around PAS was estimated to tag 15.35% of the SNP heritability for rheumatoid arthritis (7.88 fold enrichment, $p = 0.0025$). We further asked whether we could identify specific apaQTLs associated with phenotype. Indeed, 19.3% of apaQTLs (including SNPs in LD with $r^2 > 0.9$) are significantly associated with at least one trait in the UCSC GWAS catalog (Materials and methods) (*Kent et al., 2002*). For example, an apaQTL that colocalizes with the eQTLs in the C10orf88 gene (rs7904973) has been associated with increased LDL cholesterol (*Klarin et al., 2018*), suggesting that eQTLs mediated by APA can impact organismal phenotype. Taken together, we propose that APA is a complex regulatory mechanism relevant to our understanding of how genetic variation can affect disease. Thus, comprehensive maps of apaQTLs can

enhance our ability to interpret GWAS loci, particularly when the implicated variants are not eQTLs (*Joehanes et al., 2017*; *Lee et al., 2018*). For example, an apaQTL in the *ELL2* gene (rs56219066) is correlated with increased usage of an intronic PAS and is associated with risk for multiple myeloma (*Swaminathan et al., 2015*). Interestingly, multiple myeloma is among the cancer types in which widespread dysregulation of intronic APA has been documented previously (*Singh et al., 2018*; *Lee et al., 2018*).

## Discussion

Obtaining a comprehensive understanding of the mechanisms that affect gene regulation is crucial for the functional interpretation of noncoding genetic variation. Yet, existing studies that examine the role of genetic variation on APA are generally characterized by two important shortcomings. Firstly, the study of inter-individual variation in PAS usage have been mostly restricted to APA in the 3' UTRs (*Li et al., 2019*; *Yoon et al., 2012*; *Yang et al., 2020*), leaving genetic variants that impact PAS usage in other regions, for example intronic PAS, understudied. Secondly, nearly all existing studies use standard RNA-seq to estimate PAS usage, which not only limits the accuracy of usage quantification, but also makes it difficult to disentangle the contribution of co-transcriptional mechanisms to APA regulation from post-transcriptional mechanisms such as isoform-specific decay. In this study, we overcome these shortcomings by applying 3' Seq to total and nuclear mRNA fractions separately to directly measure PAS usage including that of PAS in intronic regions.

It is worthwhile to note here that despite the many advantages of using 3' Seq to identify and quantity APA isoforms, 3' Seq experiments are known to be susceptible to mispriming, which occurs when polydT primers designed to recognize the polyA-tail of transcripts anneal to adenosine stretches within a transcript, thus introducing false positive polyadenylation sites. While we used stringent criteria to reduce the effect of mispriming, we found that a small proportion of PAS used in this study may be the result of mispriming. In particular, we found an enrichment of adenosine nucleotides at a subset of intronic PAS which were discovered in our study and not previously annotated, suggesting that 10–20% of unannotated intronic PAS may be false positives (*Supplementary file 1*). To ensure that these false positive PAS do not affect the validity of our analyses, we performed the main analyses presented in this study after removing unannotated intronic PAS and found that our conclusion were robust to the small number of potential false positive intronic PAS (*Supplementary file 1*; *Wang et al., 2018*).

By collecting data from both total and nuclear mRNA fractions, we were able to study the effects of genetic variation on polyadenylation at multiple stages of the mRNA lifecycle, and to distinguish putative regulatory mechanisms by noting the stages at which the genetic effects on APA were observed. For example, genetic variants can impact steady-state isoform ratio either co-transcriptionally by affecting PAS choice during transcription (*Figure 2C*, Model 1), or post-transcriptionally by affecting binding of miRNAs or RNA-binding proteins and consequently isoform decay (*Figure 2C*, Model 2). We found that the vast majority of genetic variants that affect PAS usage ratio in total mRNA fraction, were also found to have similar effect sizes on PAS usage ratio in the nucleus. This observation implies that inter-individual variation in steady-state APA levels can generally be explained by variation in co-transcriptional mRNA processing, or mRNA processing that occur soon after transcription.

There are several co-transcriptional mechanisms that may result in variation in PAS usage. For example, previous reports have suggested that variation in the polyadenylation signal site may cause variation in PAS usage. While we found that this was the case for a small number of examples, disruption of canonical signal motifs does not appear to be a major mechanism for generating apaQTLs, an observation that is also supported by a recent study on APA in GTEx data (*Figure 2— figure supplement 5*; *Li et al., 2019*). Other possible co-transcriptional mechanisms involved in PAS choice include competition between the spliceosome and polyadenylation factors for example mediated by the spliceosomal RNA U1 (*Oh et al., 2017*), and RNAPII pausing (*Fusby et al., 2016*). Indeed, recent studies have reported that sequence and chromatin context can pause or slow down RNAPII elongation across the gene body (*Mayer et al., 2015*), suggesting that variation in RNAPII pausing may impact PAS choice (*Fusby et al., 2016*). For example, in *Drosophila melanogaster* paused RNAPII promotes the recruitment of ELAV on the pre-mRNA, which prevents usage of a proximal PAS (*Oktaba et al., 2015*). In addition, Liu et al. observed a tissue-specific shift toward

usage of proximal PAS sites in *Drosophila melanogaster* mutant for a slow elongation form of RNA-PII (*Liu et al., 2017*). These findings further suggest that variants affecting RNAPII elongation rate could underlie the genetic effects on PAS usage we detected in this study.

Although our data suggest that apaQTLs do not generally impact rates of mRNA decay, for example by affecting miRNA or RBP binding motifs, we did find some apaQTLs that appear to promote polyadenylation site choices that result in the production of isoforms with different rates of decay. For example, we observed that genetic variants that increase the usage of isoforms ending at intronic PAS tend to be associated with lower levels of gene expression. This observation is consistent with reports that isoforms with premature polyadenylation are often substrates for nonsense mediated decay or nonstop decay (*Tian and Manley, 2017*; *Vasudevan et al., 2002*). More generally, our results suggest that apaQTLs can affect gene expression levels post-transcriptionally by impacting the production of isoforms with varying levels of stability. Importantly, our study highlights APA as an eQTL mechanism independent of promoters and enhancers.

While the effect of genetic variants on gene regulation is generally assumed to move linearly from chromatin, to mRNA, to protein level, our study reveals several complex modes of genetic regulation for both gene expression and protein expression levels by APA. Although we were unable to study the genome-wide effects of APA on protein expression owing to a scarcity of protein-level data, we identified several apaQTLs that affect protein, but not gene expression levels. These results strongly suggest that APA can affect protein expression levels without affecting gene expression levels, because our power to detect genetic effects on gene expression levels far exceeds that to detect genetic effects on protein expression levels. Furthermore, some of these pQTLs were associated with ribosomal occupancy and some were not, which implies multiple pathways by which genetic variants can impact protein expression levels through APA.

In conclusion, there are many pathways through which genetic variants can impact gene regulation and, consequently, organismal phenotypes. While many studies have demonstrated the importance of gene expression regulation through promoters or enhancers, very few studies have focused on co- or post-transcriptional gene regulation. Our study shows that co- and post-transcriptional processes such as APA can mediate the effects of a substantial number of genetic variants on mRNA expression levels, protein expression levels, and risk for complex diseases.

## Materials and methods

### Cell culture

We cultured 54 Epstein-Barr virus transformed LCLs under identical conditions at 37 C and 5% $CO_2$. These LCLs were derived from Yoruba individuals originally collected as part of the HapMap project (*International HapMap Consortium, 2005*).The sampleIDs and Research Resource Identifiers (RRIDs) are the following: 184_86_N-RRID:CVCL_P452, 18486_T-RRID:CVCL_P452, 18499_N-RRID: CVCL_P457, 18499_T-RRID:CVCL_P457, 18501_N-RRID:CVCL_P458, 18501_T-RRID:CVCL_P458, 18505_N-RRID:CVCL_N803, 18505_T-RRID:CVCL_N803, 18508_N-RRID:CVCL_F279, 18508_T-RRID: CVCL_F279, 18853_N-RRID:CVCL_P470, 18853_T-RRID:CVCL_P470, 18870_N-RRID:CVCL_P479, 18870_T-RRID:CVCL_P479, 19128_N-RRID:CVCL_0X44, 19128_T-RRID:CVCL_0X44, 19141_N-RRID: CVCL_P523, 19141_T-RRID:CVCL_P523, 19193_N-RRID:CVCL_E126, 19193_T-RRID:CVCL_E126, 19209_N-RRID:CVCL_P548, 19209_T-RRID:CVCL_P548, 19223_N-RRID:CVCL_P553, 19223_T-RRID: CVCL_P553, 19225_N-RRID:CVCL_P554, 19225_T-RRID:CVCL_P554, 19238_N-RRID:CVCL_9633, 19238_T-RRID:CVCL_9633, 19239_N-RRID:CVCL_9634, 19239_T-RRID:CVCL_9634, 19257_N-RRID: CVCL_P560, 19257_T-RRID:CVCL_P560, 18511_T-RRID:CVCL_P462, 18511_N-RRID:CVCL_P462, 18519_T-RRID:CVCL_P465, 18519_N-RRID:CVCL_P465, 18520_T-RRID:CVCL_P466, 18520_N-RRID: CVCL_P466, 19092_T-RRID:CVCL_P498, 19092_N-RRID:CVCL_P498, 18858_T-RRID:CVCL_P473, 18858_N-RRID:CVCL_P473, 18861_T-RRID:CVCL_P475, 18861_N-RRID:CVCL_P475, 19119_T-RRID: CVCL_P513, 19119_N-RRID:CVCL_P513, 19130_T-RRID:CVCL_0X49, 19130_N-RRID:CVCL_0X49, 19210_T-RRID:CVCL_P549, 19210_N-RRID:CVCL_P549, 18909_T-RRID:CVCL_P487, 18909_N-RRID: CVCL_P487, 18916_T-RRID:CVCL_P492, 18916_N-RRID:CVCL_P492, 19160_T-RRID:CVCL_P533, 19160_N-RRID:CVCL_P533, 19171_T-RRID:CVCL_P534, 19171_N-RRID:CVCL_P534, 19200_T-RRID: CVCL_P543, 19200_N-RRID:CVCL_P543, 19137_T-RRID:CVCL_P520, 19137_N-RRID:CVCL_P520, 19152_T-RRID:CVCL_P530, 19152_N-RRID:CVCL_P530, 19153_T-RRID:CVCL_P531, 19153_N-RRID:

CVCL_P531, 19093_T-RRID:CVCL_P499, 19093_N-RRID:CVCL_P499, 18912_T-RRID:CVCL_P489, 18912_N-RRID:CVCL_P489, 19144_T-RRID:CVCL_P525, 19144_N-RRID:CVCL_P525, 19140_T-RRID: CVCL_P525, 19140_N-RRID:CVCL_P525, 19131_T-RRID:CVCL_P519, 19131_N-RRID:CVCL_P519, 19207_T-RRID:CVCL_P547, 19207_N-RRID:CVCL_P547, 18498_N-RRID:CVCL_P456, 18498_T-RRID: CVCL_P456, 18504_N-RRID:CVCL_P460, 18504_T-RRID:CVCL_P460, 18510_N-RRID:CVCL_P461, 18510_T-RRID:CVCL_P461, 18516_N-RRID:CVCL_P463, 18516_T-RRID:CVCL_P463, 18522_N-RRID: CVCL_P467, 18522_T-RRID:CVCL_P467, 18502_N-RRID:CVCL_P459, 18502_T-RRID:CVCL_P459, 18517_N-RRID:CVCL_P464, 18517_T-RRID:CVCL_P464, 18855_N-RRID:CVCL_P471, 18855_T-RRID: CVCL_P471, 18852_N-RRID:CVCL_P469, 18852_T-RRID:CVCL_P469, 18856_N-RRID:CVCL_P472, 18856_T-RRID:CVCL_P472, 18862_N-RRID:CVCL_P476, 18862_T-RRID:CVCL_P476, 18913_N-RRID: CVCL_P490, 18913_T-RRID:CVCL_P490, 19101_N-RRID:CVCL_P503, 19101_T-RRID:CVCL_P503, 19138_N-RRID:CVCL_P521, 19138_T-RRID:CVCL_P521, 18907_N-RRID:CVCL_P485, 18907_T-RRID: CVCL_P485. Details for each cell line are found in *Supplementary file 3*. We grew cells in a gluta-mine depleted RPMI [RPMI 1640 1X from Corning (15–040 CM)], completed with 15% FBS, 2 mM GlutaMAX (from gibco 35050–061), 100 IU/ml Penicillin, and 100 ug/ml Streptomycin. After passag-ing them 3 times the lines were maintained at a concentration of $1 \times 10^6$ cells per mL. In preparation for extraction, we allowed the cells to grow until a concentration of $1 \times 10^6$ cells per mL was reached and then proceeded to extraction.

## Collection and RNA extraction

We collected 30 million cells from each line and divided them into two 15 million cell aliquots. We spun the cells down at 500 RPM at 4C for 2 min, and then washed the pellets with phosphate-buff-ered saline (PBS) and spun down again. After this we aspirated the PBS, leaving the cell pellet. All washing steps occurred on ice or in cooled centrifuges. At this point every cell line had two separate pellets each from an input of 15 million cells. From each line we took one of these pellets for nuclear isolation. We then carried out nuclear isolation using the nuclear isolation steps outlined by *Mayer and Churchman, 2016*. Once we washed and spun down the pellets in the nuclei wash buffer, we resuspended them in 700 ul of the QIAzol lysis reagent (Qiagen). We extracted both RNA cell pellets from the same line in the same batch using the miRNeasy kit (Qiagen) according to man-ufacture instructions, including the DNase step to remove potentially contaminated genomic DNA. Details for the collection such as cell viability and cell concentration at time of collection are found in *Supplementary file 3*. We checked the quality of the collected RNA using a nanodrop. RNA concen-trations and absorbance levels from the collection are in *Supplementary file 3*.

In order to verify fraction separation, we completed the Mayer and Churchman protocol to isolate chromatin and collected cell lysates for each step in the fractionation (*Mayer and Churchman, 2016*). We performed western blots against both GAPDH (GAPDH antibody (6C5) Life Technologies AM4300) and the Carboxyl Terminal Domain of Pol-II (CTD) (Pol II CTD Ser5-P antibody, Active Motif, 61085). We ran each lysate on Mini-protean TGX precast gels (bioRad 456–1093) after digest-ing any remaining DNA molecules from the nuclear isolate with benzonase nuclease. We used Goat anti-Mouse IgG (H+L) (Invitrogen 32430) as a secondary antibody for the GAPDH antibody and Goat anti-Rat IgG (H +L) (Invitrogen 31470) as a secondary antibody for the CTD antibody. We diluted all antibodies in a 1:1000 dilution with blocking solution made from dry milk (LabScientific Lot 1267N Cat M0841). We show GAPDH isolated in the cytoplasm and CTD to the chromatin fraction (*Fig-ure 1—figure supplement 7*).

## 3' sequencing library generation

We generated 108 single-end RNA 3' sequencing libraries from the total and nuclear RNA extract using the QuantSeq 3' mRNA-Seq Library Prep Kit (*Moll et al., 2014*) as directed by the manufac-turer. We used 5 ng of each sample as input. We submitted the libraries for sequencing on the Illu-mina NextSeq5000 at the University of Chicago Genomics Core facility using single end 50 bp sequencing.

## 3' sequencing data processing

We mapped 3' Seq reads to hg19 (*Church et al., 2011*) using STAR RNA-seq aligner (*Dobin et al., 2013*) using default settings with the WASP mode to filter out reads mapping with allelic bias

(*van de Geijn et al., 2015*). Similar to previously published 3' Seq methods, we accounted for internal priming by filtering reads preceded by 6 Ts in a row or 7 of 10 Ts in the 10 bases directly upstream of the mapping position in the reference genome (*Tian et al., 2005*; *Sheppard et al., 2013*; *Beaudoing et al., 2000*). We verified the individual identity of all bam files using VerifyBamID (*Jun et al., 2012*). Due to low confidence in the identity of 2 individuals, they were removed from all analysis. Raw read and mapped read statistics after accounting for internal priming can be found in *Supplementary file 3* (*Figure 1—figure supplements 8–11*).

## Identification and characerization of PAS

We merged all mapped reads and called peaks using an inclusive method, identifying all regions of the genome with non-zero read counts in 90% percent of libraries and an average read count of greater than 2 counts. This resulted in 138,181 peaks. We assigned each of these peaks to a genic location according to NCBI Refseq annotations for 5' UTRS, 3' UTRs, exons, introns, and regions 5 kb downstream of annotated genes downloaded from the UCSC table browser (*Kent et al., 2002*). When a region mapped to multiple genes we used a hierarchical model, similar to the method used by *Lin et al., 2012* to assign the peak to a gene annotations. Our method prioritizes annotations in the following order: 3' UTRs, 5 kb downstream of genes, exons, 5' UTRs, and introns. To further verify absence of PAS detected as a result of internal priming we removed PAS with 6A's or 70% As in the 15 basepairs downstream of the site. We next utilized a gene level noise filter to account for non-uniform read coverage across the genome. We created a usage score for each PAS based on of the number of reads mapping to the PAS over the number of reads mapping to any PAS associated with the same gene. We filtered out peaks with a mean usage of less than 5% in both the total and nuclear libraries. After this filter, we were left with 35,032 PAS in the total mRNA fraction and 39,164 PAS in the nuclear fraction. The merged set with PAS from both fractions used for PAS QC is available on GEO and has 41,810 PAS. We compared our set of PAS to the human PolyADB release 3.2 annotation (*Wang et al., 2018*; *Figure 1—figure supplement 6*). We explored the relationship between number of PAS detected and gene expression using TPM estimates from YRI LCLs after removing very lowly expressed genes (less than 1 TPM) (*Lappalainen et al., 2013*). We calculated the 5' splice site strength using the MaxEntScore tool, for each of the introns in our annotation (*Yeo and Burge, 2004*). We binned the introns by decile according to the scores and evaluated the distribution of the introns containing PAS. We also used the scores for the introns containing PAS to investigate the relationship between PAS usage and 5' splice site strength.

## PAS signal site enrichment and locations

We used the Homer findMotifsGenome.pl script with the -size −300,100 option to identify binding motifs in the 50 bp upstream of each PAS (*Heinz et al., 2010*). As a background, we used genome shuffle to randomly chose the same number of 50 bp regions. To explore the location of the signal site relative to the PAS (most 3' end of each identified peak), we determined the relative position of previously described potential signal sites to this position (*Beaudoing et al., 2000*). We then extended each PAS 100 bp upstream and identified the starting position of each of the 12 PAS signal site variations identified by Beaudoing et al. without allowing for sequence mismatch (*Beaudoing et al., 2000*).

## Differential isoform analysis

We mapped 3' Seq reads to all PAS peaks with mean coverage of 5% in the total or nuclear fraction libraries. This results in 41,813 annotated sites. We assigned reads to PAS using the featureCounts tool with the -O flag to assign reads to all overlapping features (*Liao et al., 2014*). We ran the leafcutter_ds.R script on chromosomes 1–22 separately using the cellular fraction label as the sample group identifier (*Li et al., 2018*). This analysis tests 9790 genes and resulted in 8227 genes with significant (FDR 10%) isoform level differences between the total and nuclear cellular fraction. We called differentially used PAS as sites with a Δ polyadenylation site usage (ΔPAU) greater than 0.2 or less than −0.2. In our analysis a positive ΔPAU corresponds to increase usage in the total cellular fraction while a negative ΔPAU corresponds to increased usage in the nuclear fraction.

## apaQTL calling in both fractions

We used the Leafcutter prepare_phenotype_table.py script with default settings to normalize the PAS usage ratios across individuals within each fraction. This method also outputs the top principal components (PCs) of the data to use as covariates. We plotted the proportion of variation explained by each PC in order to identify the number of PCs to include in the analysis (*Figure 2—figure supplement 3*). We included the top 4 PCs as well as the library preparation batch as the covariates. We plotted the proportion of variance explained by a number of cofactors in each of the top 10 PCs (*Figure 2—figure supplement 3*). The top four PCs correlate most strongly with the cell count at collection (*Figure 2—figure supplement 3*). We used the same genotypes from *Li et al., 2016*, available at http://eqtl.uchicago.edu/jointLCL/genotypesYRI.gen.txt.gz (*Li et al., 2016*). We removed individual NA19092 due to lack of genotype information in this file, bringing our sample size to 51 individuals for this part of the analysis. Only SNPs with a MAF >5% in our sample were included. We used FastQTL to map apaQTLs in cis (25 kb on either side) with 1000 permutations to select the top SNP-PAS association (*Ongen et al., 2016*). We called apaQTLs in each fraction as variants passing 10% FDR (Benjamini-Hockberg) after permutations. In order to plot interpretable effect sizes for each association we computed nominal PAS:SNP associations for the pre-normalized PAS ratios.

## Association of apaQTLs with chromatin states

We downloaded the GM12878 chromatin HMM annotations for hg19 from the UCSC table browser (*Kent et al., 2002*). We overlapped the eQTLs identified and published in *Li et al., 2016* as well as the total and nuclear fraction apaQTLs with these categories. We calculated 95% confidence intervals for each measurement by sampling the number of QTLs in the set with replacement 1000 times (*Figure 3—figure supplement 3*).

## apaQTL overlap with eQTLs

We obtained the set of explained and unexplained eQTLs from *Li et al., 2016*. In order to test whether genes with an unexplained eQTL are more likely to be explained by variation in APA, we separated the permuted apaQTL association (top snp per PAS) into three categories: unexplained eGene, explained eGene, non eGenes. We tested for significant enrichment of apaQTLs in each category using one-sided Wilcoxon rank sum tests. In order to test if each explained and unexplained eQTLs described in *Li et al., 2016* overlaps with an apaQTL, we extracted the nominal associations for each eQTL gene-SNP pair from the apaQTL data in both fractions. In order to account for multiple PAS associations for each pair, we selected the most significant p-value and used a Bonferroni correction to account for the number of PAS tested in the gene. We consider an eQTL as explained by an apaQTL if the corrected p-value is less than 0.05 but report the values for a range of cutoffs in *Figure 3—figure supplement 4*). We performed colocalization with the R coloc package (*Wallace et al., 2012*). The Bayes Factor colocalization method reports Bayes Factors for 4 alternative hypotheses. PP0: No association with either trait, PP1: No association with trait 1, PP2: No association with trait 2, PP3: Association with trait 1 and trait 2, two independent SNPs, and PP4: Association with trait 1 and trait 2, one shared SNP. If causal SNPs for an apaQTL and an eQTL is the same SNP, then PP4 is expected to be large (greater than 0.5). We accounted for incomplete power using the method described in Ongen et al. (*Supplementary file 1*; *Ongen et al., 2017*).

## apaQTLs overlap with ribosome specific and protein specific QTLs

The list of protein specific QTL genes can be found in the supplementary information from *Battle et al., 2015*. In order to show that genes with an eQTL and protein specific QTLs are likely to be associated with APA, we separated the permuted apaQTL association (top snp per PAS) into three categories: eGene, pGene, or neither pGene nor eGene. We performed the same analysis with rGenes, eGenes, and neither rGenes nor eGenes. We tested for significant enrichment with one sided Wilcoxon rank sum tests (*Figure 4A* and *Figure 4—figure supplement 1*).

## Identification of molecular QTL associations

We sought to test if SNPs identified as apaQTLs are significantly associated with other molecular phenotypes previously tested in the same panel of LCLs. We tested for associations between the

genotypes used in this study and each gene for each phenotype with fastqtl using the top 5 PCs calculated in *Li et al., 2016* as covariates. We used normalized RNA expression, RiboSeq values, and protein levels, published in *Li et al., 2016*.

## PAS heritability estimates and apaQTL overlap with GWAS catalog

We downloaded GWAS summary statistics from both *Astle et al., 2016*; *Okada et al., 2014* We augmented our PAS sites by 500 bp on either side and ran LD score regression using methods described in *Bulik-Sullivan et al., 2015* We downloaded the CRCh37hg19 GWAS catalog for UCSC table browser (*Kent et al., 2002*). We identified SNPs in LD with the nuclear apaQTLs using the LDproxy tool from LDlink with YRI as the population (*Machiela and Chanock, 2015*). We filtered all results to SNPs with an r2 greater than 0.9. We overlapped the full set with the GWAS catalog using pybedtools.

## Acknowledgements

We thank N Gonzalez, JP Staley, MC Ward for comments on the manuscript. Funding: This work was supported by the US National Institutes of Health (R01GM130738 to YIL). BEM supported by T32 GM09197 to the University of Chicago and F31HL149259 to BEM from National Heart, Lung, And Blood Institute of the National Institutes of Health. SP was in part supported by the National Center for Advancing Translational Sciences of the NIH (K12 HL119995). This work was completed in part with resources provided by the University of Chicago Research Computing Center.

## Additional information

### Funding

| Funder | Grant reference number | Author |
|---|---|---|
| National Institutes of Health | T32 GM09197 | Briana E Mittleman |
| National Institutes of Health | F31HL149259 | Briana E Mittleman |
| National Institutes of Health | R01GM130738 | Yang Li |
| National Institutes of Health | K12 HL119995 | Sebastian Pott |

The funders had no role in study design, data collection and interpretation, or the decision to submit the work for publication.

### Author contributions

Briana E Mittleman, Data curation, Formal analysis, Funding acquisition, Visualization, Methodology, Writing - original draft, Writing - review and editing; Sebastian Pott, Conceptualization, Supervision, Funding acquisition, Writing - review and editing; Shane Warland, Data curation, Writing - review and editing; Tony Zeng, Formal analysis, Visualization; Zepeng Mu, Formal analysis, analysis and interpretation of data; Mayher Kaur, Visualization, analysis and interpretation of data; Yoav Gilad, Conceptualization, Resources, Supervision, Writing - review and editing; Yang Li, Conceptualization, Resources, Formal analysis, Supervision, Funding acquisition, Writing - review and editing

### Author ORCIDs

Briana E Mittleman https://orcid.org/0000-0002-4979-4652
Sebastian Pott http://orcid.org/0000-0002-4118-6150
Zepeng Mu https://orcid.org/0000-0002-7717-3247
Yoav Gilad http://orcid.org/0000-0001-8284-8926
Yang Li https://orcid.org/0000-0002-0736-251X

### Decision letter and Author response

Decision letter https://doi.org/10.7554/eLife.57492.sa1
Author response https://doi.org/10.7554/eLife.57492.sa2

## Additional files

### Supplementary files

• Supplementary file 1. Supplementary Text.

• Supplementary file 2. ApaQTL whose lead SNP is nominally associated with protein expression levels but not expression. Table includes p-value and slope for the associated between the lead SNP and nuclear APA usage, gene expression levels, protein expression levels, and ribosome occupancy (as measured using ribo-seq).

• Supplementary file 3. Library information for each Yoruba lymphoblastoid cell line, including sample, collection, and read information.

• Transparent reporting form

### Data availability

Fastq files and PAS annotations are available at GEO under accession GSE138197. All reproducible scripts and software versions can be found at through Zenodo with https://doi.org/10.5281/zenodo.3905372.

The following datasets were generated:

| Author(s) | Year | Dataset title | Dataset URL | Database and Identifier |
|---|---|---|---|---|
| Mittleman BE, Pott S, Warland SF, Zheng T, Mu Z, Kaur M, Gilad Y, Li YI | 2020 | Alternative polyadenylation mediates genetic regulation of gene expression | https://www.ncbi.nlm.nih.gov/geo/query/acc.cgi?acc=GSE138197 | NCBI Gene Expression Omnibus, GSE138197 |
| Mittleman BE, Pott S, Warland SF, Zheng T, Mu Z, Kaur M, Gilad Y, Li YI | 2020 | Alternative polyadenylation mediates genetic regulation of gene expression | https://doi.org/10.5281/zenodo.3905372 | Zenodo, 10.5281/zenodo.3905372 |

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
