## [Decision Letter]

**Acceptance summary:**

This article presents an interesting link between alternative polyadenylation, decay rates and genetic variation in gene expression control.

**Decision letter after peer review:**

[Editors’ note: the authors submitted for reconsideration following the decision after peer review. What follows is the decision letter after the first round of review.]

Thank you for submitting your work entitled "Alternative polyadenylation mediates genetic regulation of gene expression" for consideration by *eLife*. Your article has been reviewed by three peer reviewers, one of whom is a member of our Board of Reviewing Editors, and the evaluation has been overseen by a Senior Editor. The reviewers have opted to remain anonymous.

Our decision has been reached after consultation between the reviewers. Based on these discussions and the individual reviews below, we regret to inform you that your work cannot be considered further for publication in *eLife*, at least in its current form. However, a substantially revised version that successfully addresses all the major comments raised in the reviews below would be suitable for reconsideration.

Li and colleagues perform 3'seq on nuclear and total RNA to evaluate the diversity of alternative polyadenylation (APA) site choice from 52 lymphoblastoid cell lines. A surprising finding that is a large fraction of the nuclear fraction apaQTLs are intronic. This suggests that apaQTLs may contribute to eQTLs that are not associated with chromatin fractions and can lead to protein level changes without RNA abundance differences. A novel mechanism for genetic variants that modify ribosome occupancy of a transcript independently of its expression level is suggested. While the manuscript is certainly intriguing and highlights the potential importance of APA to human biology, which is of interest to the RNA and genetics field, there are many significant technical concerns that need reanalysis and clarification, in order for the biological implications to be sufficiently justified. The final results are somewhat underwhelming and in its current form insufficient to cross the threshold for publication in *eLife*.

Reviewer #1:

In this manuscript, Mittleman and colleagues contribute another layer of annotations to a historied collection of LCLs. By performing 3' Seq on nuclear and total mRNA, the authors are able to document the diversity of polyadenylation site choice in cells and the extensive regulatory activity occurring in between nascent transcription and steady state mRNA levels. They find that polyadenylation sites that produce nonfunctional transcripts (predominately but not solely in intronic segments) are invisible in standard RNA-seq datasets. Although they may be used uncommonly, these noncanonical sites may outnumber those in 3' UTRs. The authors extend the repertoire of QTL calling and learn that apaQTLs exhibit chromatin and other features distinct from those of eQTLs. More importantly, they find a novel mechanism for genetic variants that modify ribosome occupancy of a transcript independent of its expression level. However, while this work presents a transparently useful resource, the manuscript is conspicuously narrow in scope; it doubles down on alternate polyadenylation site architecture and chromatin state even when the identified trends are subtle at best.

1) The main takeaways (polyadenylation site choice is more diverse than expected, and traditional datasets obscure the role of alternative polyadenylation in mediating eQTLs) are cemented by the schematic in Figure 4C, but startlingly few figures address this point elsewhere. Figure 2D and Figure 3—figure supplement 3 and 4 are offered as supporting evidence, but the entirety of at least one main text figure should be dedicated to this in this reviewer's opinion. The principal findings simply need more attention.

Figure 1B elliptically addresses mRNA stability versus nucleus-biased transcripts, but there is a lot to unpack here. The panel title argues that polyadenylation choice is only weakly correlated, but the text takes care to establish a positive correlation. The trendline looks very ill fit to me. This is a rather critical point: is 3' Seq complementary, redundant or synergistic with 4su-seq? In other words, is 3' Seq really telling us something new, or is it recapitulating 4su? Additionally, Figure 2D is presented as critical evidence, and I feel that more should be done to bolster it. I was under the impression that more apaQTLs were called in nuclear than total mRNA fractions because transcripts had undergone decay. If effect sizes aren't changing, then does that mean both alleles of apaQTLs are equally disenriched in total mRNA fractions? The discussion does not take a clear stance on the relationship between alternative polyadenylation, translational efficiency and mRNA decay, which should be central to the message of the paper.

2) The majority of the plots delve into subtle differences of polyadenylation in various transcript elements (like introns and UTRs) and in nuclear mRNA versus total mRNA and for explained or unexplained QTLs. While these figure panels may show some slight differences (e.g. Figure 3), this leaves the reader with questions.

What percent of nonsense-mediated decay is dedicated to screening prematurely polyadenylated transcripts? For simultaneous apaQTL/eQTLs, approximately what percent of eQTL variance could be explained by apaQTL variance? Is the same extent of intronic alternative polyadenylation in nuclei observed in other cell types? Does higher variance in alternative polyadenylation for a transcript correlate with higher variance in ribosome occupancy (irrespective of QTLs)? Are alternatively polyadenylated sites enriched for heritability for a panel of traits (i.e. using LD score regression)? More conserved? Are genes with more frequent alternative polyadenylation longer? More highly expressed? Expressed more selectively across tissues? Enriched for certain gene annotations (e.g. GO terms)? More likely to contain certain RNA-binding protein motifs? More likely to bind RBPs as shown by CLIP? Less likely to be annotated with multiple transcription start sites?

3) The manuscript is not an easy read. Even with the current results, the format and layout would be daunting for generalists unfamiliar with this subject matter. Figures are called out too soon in some cases. Axis and panel titles frequently obfuscate rather than illuminate the subject matter. Numerous tracks are illegible at true size and somewhat difficult to discern even in the zoomed in figures (nonetheless a very helpful addition to the manuscript). These points really do need to be addressed.

4) The content of Figure 4 is quite anecdotal. Ideally there would be follow up work on the mechanism of riboQTLs or pQTLs without eQTLs. It may also be appropriate to combine Figures 3 and 4 and send some panels to the supplementary materials.

Reviewer #2:

In this paper, Mittleman et al. used 3' seq to analyze alternative polyadenylation (APA) isoform expression levels in nuclear and total RNA fractions from 52 lymphoblatoid cell lines. They identified about 600 apaQTLs in both fractions. A surprising finding is that a large portion of the nuclear fraction apaQTLs are intronic. They indicate that apaQTLs may contribute to some eQTLs that have not been associated with chromatin functions and can also lead to protein level changes without RNA abundance difference. Overall this proof-of-principle type work highlights the functional relevance of APA to human traits, a message that is of value to both RNA processing and human genetics fields. The experiment and data analysis were well carried out in general. However, some aspects of this work need further polishing and alternative explanations need to be considered in their data interpretations.

It is not clear to what extent internal priming cases have been addressed. The QuantSeq kit uses oligo(dT) for RT priming, which can lead to substantial internal priming at A-rich regions of RNA. Even though the authors seem to have employed a rigorous computational approach to cull their data, internal priming cases can still exist. One way to gauge the extent is to check the nucleotide frequency profile around the polyA sites that matched polyA DB vs. those did not. If internal priming problem persists, they would see an A-rich peak around the polyA site for those non-matched sites. This issue is highly relevant to their conclusion, because many of the intronic polyA sites could well be A-rich regions in retained introns. As such, some of the cases might in fact be intron retention rather than intronic polyadenylation.

For intronic polyA regulation, the authors need to consider the possibility of variations of 5' splice site strength and/or intron size (through insertion or deletion), which were shown to be important for intronic polyadenylation by Tian et al., 2007.

The authors claim that the relative number of nuclear seq reads to total reads is indicative of RNA decay. This is not well supported by their data. The data shown in Figure 1—figure supplement 1 had a quite dismal correlation coefficient and the p-value is not 2.2x10-16 as mentioned in the main text. The possibility of nuclear export control, in addition to decay, should be considered.

The description of ribosome occupancy is quite scant. Because intronic polyadenylation would truncate transcripts, change of ribosome occupancy could be simply due to transcript size change (thus ribosome number per nucleotide changes) rather than ribosome number per transcript. The authors need to distinguish these two different scenarios.

Reviewer #3:

The authors are studying genetic variation that affect APA. While there has been a fair amount of interest in the study of eQTLs the focus on this specific mechanism is less explored. This summer there was a fairly large study of genetic variation and apa published this summer (PMID: 31475030). It may be appropriate to mention this work to set the study in the appropriate context. Nevertheless, there are novel contributions in this study that offer more resolution in terms of mechanism. The work uses 3' RNA-seq (3' Seq) to measure PAS usage in whole cells as well as the nucleus for the purpose of distinguishing differences in PAS usage from differential stability. The use of lymphoblastoid cell lines was a good decision given the deep characterization of this resource in the literature. In general, the study is thorough and makes a solid contribution. I felt that the all the computational analyses were thorough, appropriate and executed well. On the negative side, there is a potential limitation that the functional consequence of the events cannot really be determined from the relative RNA-seq measures in nuclear versus whole cell. It is possible the eQTL that trigger APA create changes in isoform ratios but these are not reflected in ribosome/polysome associated transcripts. In general, I struggled to identify any single high impact discovery but in its totality there is significant biology in the paper in terms of the discovering how variants expressed gene expression.

In terms of impact and broad appeal, I would think this would be appropriate for *eLife*.

---

## [Author Response]

[Editors’ note: the authors resubmitted a revised version of the paper for consideration. What follows is the authors’ response to the first round of review.]

Li and colleagues perform 3'seq on nuclear and total RNA to evaluate the diversity of alternative polyadenylation (APA) site choice from 52 lymphoblastoid cell lines. A surprising finding that is a large fraction of the nuclear fraction apaQTLs are intronic. This suggests that apaQTLs may contribute to eQTLs that are not associated with chromatin fractions and can lead to protein level changes without RNA abundance differences. A novel mechanism for genetic variants that modify ribosome occupancy of a transcript independently of its expression level is suggested. While the manuscript is certainly intriguing and highlights the potential importance of APA to human biology, which is of interest to the RNA and genetics field, there are many significant technical concerns that need reanalysis and clarification, in order for the biological implications to be sufficiently justified. The final results are somewhat underwhelming and in its current form insufficient to cross the threshold for publication in eLife.

Thank you for sending the reviews and allowing us to submit a revised version. We also thank all reviewers for their help in improving this manuscript.

We have responded to each of the reviewer’s concerns and suggestions, as detailed below, and more generally, we have completely restructured the figures and edited the text to address the general sentiment that the main insights in our paper were not clearly reported.

Of particular note:

Reviewer 1 challenged us with a large number of questions, and we have carried out all of the proposed additional analyses. We want to particularly thank this reviewer, as we believe that our reported study is better and more robust with these additional insights.

Reviewer 2 challenged us to further explore the potential issue of mispriming, and we are grateful. We believe that our analysis is now much improved.

Reviewer #1:1) The main takeaways (polyadenylation site choice is more diverse than expected, and traditional datasets obscure the role of alternative polyadenylation in mediating eQTLs) are cemented by the schematic in Figure 4C, but startlingly few figures address this point elsewhere. Figure 2D and Figure 3—figure supplement 3 and 4 are offered as supporting evidence, but the entirety of at least one main text figure should be dedicated to this in this reviewer's opinion. The principal findings simply need more attention.

Thank you for the comment. We have now completely revised the structure of the figures and several sections of the manuscript to improve the focus on the main claims.

Figure 1B elliptically addresses mRNA stability versus nucleus-biased transcripts, but there is a lot to unpack here. The panel title argues that polyadenylation choice is only weakly correlated, but the text takes care to establish a positive correlation. The trendline looks very ill fit to me. This is a rather critical point: is 3' Seq complementary, redundant or synergistic with 4su-seq? In other words, is 3' Seq really telling us something new, or is it recapitulating 4su?

We are sorry for this misunderstanding. We initially included Figure 1B to show that 3’-Seq collected from the nuclear cell fraction represents mRNA that are less subject to decay. This was meant to be a quality control and not speak to whether 3’-Seq is complementary, redundant, nor synergistic with 4sU. In other words, we simply wanted to highlight that nuclear mRNA 3’-Seq and total mRNA 3’-Seq provide different information, in case one might ask whether the two data sets are redundant.

We have now moved Figure 1B to S as it simply represents a quality control that we performed.

Additionally, Figure 2D is presented as critical evidence, and I feel that more should be done to bolster it. I was under the impression that more apaQTLs were called in nuclear than total mRNA fractions because transcripts had undergone decay. If effect sizes aren't changing, then does that mean both alleles of apaQTLs are equally disenriched in total mRNA fractions? The discussion does not take a clear stance on the relationship between alternative polyadenylation, translational efficiency and mRNA decay, which should be central to the message of the paper.

We thank the reviewer for asking this question, and again we are sorry for the confusion. As mentioned above, the compassion with the RNA decay data was for reported simply as a QC measure and we have now moved it to the supplement.

We also now clarify that most of the apaQTLs are discovered in both fractions and have highly correlated effect sizes. The implication of this observation is that **most** genetic variants impact steady-state relative isoform usage during transcription.

We also revised the text to clarify the following results: We show that only a subset of apaQTLs impact gene expression levels by increasing the usage of isoforms with lower stability. We also show that a subset of apaQTLs impact translational efficiency, and protein expression levels, independently of effects on mRNA expression levels.

2) The majority of the plots delve into subtle differences of polyadenylation in various transcript elements (like introns and UTRs) and in nuclear mRNA versus total mRNA and for explained or unexplained QTLs. While these figure panels may show some slight differences (e.g. Figure 3), this leaves the reader with questions.

Thank you for these questions. We hope our explanations below address these concerns. We added many of the analyses described below to the paper.

What percent of nonsense-mediated decay is dedicated to screening prematurely polyadenylated transcripts?

It is very difficult to answer this question using our 3’-seq data as our data does not capture nonsense-mediated decay transcripts. We can only infer decay (or lack of export) by observing differences between nuclear and total mRNA fractions.

That said, to gain insights into the links between NMD and APA, we considered a list of genes known to be a target of NMD from a study conducted by Colombo et al. (Colombo et al., 2017). Interestingly, we found that genes with differential usage between the total and nuclear fractions are significantly enriched among genes targeted by NMD (2.14X enrichment, p-value=4.59x10^-5^).

We also found that the 57 examples of QTLs with larger effect sizes in nuclear than in total are significantly enriched for genes targeted by NMD (p=0.0017, hypergeometric test). This observation suggests that the genetic effects on gene expression through APA may be mediated through NMD.

One caveat of our analysis is that the Colombo et al. study used Hela cells rather than LCLs. We have thus decided to omit this result from the revised manuscript.

For simultaneous apaQTL/eQTLs, approximately what percent of eQTL variance could be explained by apaQTL variance?

Good questions! We cannot address it by directly comparing the effect sizes because the effect sizes for apaQTLs and eQTLs are not on the same scale (the APA phenotype is a ratio of reads while expression is a sum of reads across a gene). Instead, to address this question, we considered colocalization of the eQTLs and apaQTLs. If the eQTL and apaQTL colocalize (are likely to have the same causal SNP), we can infer that the apaQTL explains 100% of the eQTL variance.

To quantify the amount of colocalization between our apaQTLs and eQTLs, we used coloc (Wallace et al., 2012) package to test for whether the phenotypes share a causal SNP. We found that eQTLs and apaQTLs in the vast majority of genes (78.8%) are more likely to share a causal SNP than not. Thus, we conclude that most apaQTLs that are determined to be eQTLs are likely to be causal, and thus likely explain most or all the SNP effect on gene expression.

These results are now reported in the paper in subsection “Impact of apaQTLs on gene expression levels”.

Is the same extent of intronic alternative polyadenylation in nuclei observed in other cell types?

To the best of our knowledge, we are the first study to separate nuclei from whole cells with the goal to quantify intronic APA, so we are unable to provide a comprehensive answer to this question without collecting much more data.

Nevertheless, to provide some insight, we reasoned that because total mRNA captures a small fraction of nuclear mRNA, it may be possible to use total mRNA to quantify the extent of intronic alternative polyadenylation in nuclei. For example, we found that 387 intronic PAS that were highly used in LCL nuclear mRNA were also detectable in LCL total mRNA. We can thus ask what fraction of these 387 intronic PAS also show evidence of usage in other cell-types from data collected by other studies on PAS, which used a similar strategy for collecting total polyadenylated mRNA to ours.

As baseline, we used 3’-seq usage data collected by Lianoglou et al., which include LCLs and four other cell-types (Breast, Ovary, Testes, Stem Cells). We found that about 10% of the 387 intronic PAS showed detectable usage in total 3’ seq from LCLs collected by the Lianoglous study. By contrast, around 5% of the intronic PAS showed usage in Breast, and Testes. Usage of 3’ Seq data from another study performed by Derti and colleagues suggest that nearly 10% of the 387 PAS showed detectable usage.

Put together, these results suggest that there is at most a 2-fold difference in alternative polyadenylation in nuclei in other cell-types.

These results are now reported in the paper in subsection “Alternative polyadenylation in human LCLs as defined using Nuclear and Total mRNA 3’ Seq”.

Does higher variance in alternative polyadenylation for a transcript correlate with higher variance in ribosome occupancy (irrespective of QTLs)?

There are multiple polyadenylation sites per gene, by definition, but only one measure of ribosome occupancy per gene. To overcome this difficulty, we simply asked whether the variance of the usage of the most highly used PAS correlates with variance in occupancy. We observed a small but significant positive correlation between individual variance in APA usage and ribosome occupancy (Correlation = 0.15, p <2.2x10^-16^).

These results are now reported in the paper in subsection “APA mediates gene regulation independently of mRNA expression levels”.

Are alternatively polyadenylated sites enriched for heritability for a panel of traits (i.e. using LD score regression)?

This is a very good idea, thank you. We addressed this question by considering LD score regression with 1kb regions around each PAS and a panel of immune related traits. Using our PAS augmented by 500bp on each site as annotation, we observed a significant enrichment for many traits, notably rheumatoid arthritis, lymphocyte count, and neutrophil count. This result strengthens our claim that genetic control of alternative polyadenylation is likely relevant for complex human phenotypes.

These results are now reported in the paper in subsection “APA mediates genetic effects on complex traits” and in Figure 4—figure supplement 3.

More conserved?

We tested whether the 50bp upstream of each PAS were more conserved than a control region of further 50bp upstream of that. We measured conservation using PhyloP scores downloaded from UCSC table browser. PhyloP scores measures conservation across 100 vertebrate species at a base pair resolution with a larger value reflecting more conservation than expected by genetic drift. For genes with one to four PAS, we saw higher phyloP scores for PAS than control regions (See Author response image 1). The difference is more pronounced the fewer PAS are identified in each gene, suggesting less evolutionary constraint on PAS when the number of PAS is large.

Are genes with more frequent alternative polyadenylation longer?

We tested this by comparing the number of identified PAS and the length of each gene. We found that genes with multiple PAS are significantly longer than genes with only 1 identified PAS (median 11.3kb vs 34.4kb, Wilcoxon p-value < 2.2x10^-16^). We also found that 3’ UTR length was highly correlated with the number of PAS in the 3’ UTR (correlation = 0.42, pvalue =< 2.2x10^-16^) (See Author response image 2).

**Author response image 2. respfig2:** 

More highly expressed?

We considered the relationship between number of identified PAS and mean normalized mRNA expression level in YRI LCLs. We do not expect a high positive correlation in this analysis because we do not expect PAS number to scale with expression. Indeed, a strong relationship between gene expression and PAS may suggest that our identification is biased towards highly expressed genes. Reassuringly, we found that there is a slight negative correlation between number of PAS and TPM after removing the very lowly expressed genes (<1 TPM) (Pearson Correlation = -0.12, p-value < 2.2x10^-16^).

These results are now reported in the paper in subsection “Alternative polyadenylation in human LCLs as defined using Nuclear and Total mRNA 3’ Seq” and in Figure 1—figure supplement 1.

Expressed more selectively across tissues?

We note that this question has previously been addressed in a 2013 studies from Lianogluo et al.(Lianoglou et al., 2013) in which genes ubiquitously expressed were found to be more likely to harbor multiple PAS. In our case, we now asked whether the number of PAS for a gene is negatively correlated with its gene expression variance from GTEx, a measure of expression selectivity. If genes that are ubiquitously expressed are more like to harbor multiple PAS, as suggested by Lianogluo and colleagues, then we would expect a negative correlation between gene expression variance across tissues and number of PAS (lower variance likely indicate ubiquitous expression). Indeed, we found a significant negative correlation between a gene’s expression variance and its number of PAS (Pearson’s correlation -0.17, p < 2.2 x 10^-16^)

**Author response image 3. respfig3:** 

Enriched for certain gene annotations (e.g. GO terms)?

We performed a GO analysis for genes with APA and identified many significant processes, functions, and components at q < 10^-9^.

More likely to contain certain RNA-binding protein motifs? More likely to bind RBPs as shown by CLIP?

To address this question, we downloaded eCLIP data for 25 different RNA binding proteins in K562 cells from the ENCODE project. We asked whether 3’ UTR with an apaQTL were more likely to be bound by an RBP as shown by eCLIP than expected by chance. Interestingly, we found that the RNA binding proteins with the strongest enrichments are FUS and SAFB. These are intriguing result given the known function of FUS as a splice factor that guide nuclear export.

These results are now reported in the paper in subsection “Genetic loci associated with variation in APA”.

Less likely to be annotated with multiple transcription start sites?

To answer this question, we downloaded CAGE-seq data performed on the nuclear fraction of GM12878 LCL, from the ENCODE consortium. We identified 9,896 genes that were measured in both CAGE-seq and 3’ Seq, with multiple PAS and multiple TSS. We found no correlation between the number of PAS and the number of TSS. However, we found that genes with multiple annotated PAS are significantly enriched for genes with multiple TSS (p = 7.97x10^-6^) compared to genes with a single TSS. It also appears like gene length is an indicator for variation in number of PAS and number of TSS (nPAS and nTSS)

**Author response image 4. respfig4:** 

3) The manuscript is not an easy read. Even with the current results, the format and layout would be daunting for generalists unfamiliar with this subject matter. Figures are called out too soon in some cases. Axis and panel titles frequently obfuscate rather than illuminate the subject matter. Numerous tracks are illegible at true size and somewhat difficult to discern even in the zoomed in figures (nonetheless a very helpful addition to the manuscript). These points really do need to be addressed.

We thank this reviewer for these criticisms, which have prompted us to revamp all figures improving flow. We have modified the figures to increase track size and to increase general readability of the paper.

4) The content of Figure 4 is quite anecdotal. Ideally there would be follow up work on the mechanism of riboQTLs or pQTLs without eQTLs. It may also be appropriate to combine Figures 3 and 4 and send some panels to the supplementary materials.

While we agree with the reviewer that we present anecdotal evidence to support our claims, it should be noted that the identification of other types of QTLs (pQTLs, riboQTLs) would require a completely new data collection and that kind of follow up work needs to be reserved to a different project. Without additional data, we now discuss proposed potentially mechanisms in subsections “Impact of apaQTLs on gene expression levels” and “APA mediates genetic effects on complex traits”.

Reviewer #2:It is not clear to what extent internal priming cases have been addressed. The QuantSeq kit uses oligo(dT) for RT priming, which can lead to substantial internal priming at A-rich regions of RNA. Even though the authors seem to have employed a rigorous computational approach to cull their data, internal priming cases can still exist. One way to gauge the extent is to check the nucleotide frequency profile around the polyA sites that matched polyA DB vs. those did not. If internal priming problem persists, they would see an A-rich peak around the polyA site for those non-matched sites. This issue is highly relevant to their conclusion, because many of the intronic polyA sites could well be A-rich regions in retained introns. As such, some of the cases might in fact be intron retention rather than intronic polyadenylation.

Thank you for this important comment. We originally took various measures to ensure that misprimed reads are not included in our analysis. For example, we include filters both at the read and PAS level according to previous reports using the same experimental protocol. Now, upon your suggestion, we considered the base composition around our PAS. We have separated PAS based on their location and whether the PAS is annotated in polyADB (Wang et al., 2018).

We found a very similar basepair composition for all PAS except for intronic PAS that are unannotated in polyA DB. This suggests some amount of mispriming as hypothesized by the reviewer. By quantifying the increase in A at nearby position around unannotated intronic PAS relative to annotated intronic PAS, we estimate that up to 20% of our unannotated intronic PAS may be explained by mispriming.

We thus repeated our analysis after removing all intronic PAS that have not been previously annotated, and we show that our results are robust with respect to this property. We report this analysis in the paper and we sincerely thank the reviewer for this suggestion.

For intronic polyA regulation, the authors need to consider the possibility of variations of 5' splice site strength and/or intron size (through insertion or deletion), which were shown to be important for intronic polyadenylation by Tian et al., 2007.

We thank the reviewer for this suggestion. Tian et al. reported that introns with intronic PAS are associated with weak 5’ splice sites (SS). We used MaxEntScan scores as proxy for the 5’ SS strength for all of our nuclear intronic PAS (Yeo and Burge, 2004). We found that, as expected, the top 10% most highly used intronic PAS are in introns with significantly lower maxent scores than the bottom 10% (6.43 vs 7.26, Wilcoxon p=1.4x10^-3^). However, we found no a correlation between mean PAS usage and the 5’ SS strength. We added these results to subsection “Alternative polyadenylation in human LCLs as defined using Nuclear and Total mRNA 3’ Seq”.

The authors claim that the relative number of nuclear seq reads to total reads is indicative of RNA decay. This is not well supported by their data. The data shown in Supplementary Figure 1 had a quite dismal correlation coefficient and the p-value is not 2.2x10-16 as mentioned in the main text. The possibility of nuclear export control, in addition to decay, should be considered.

We are sorry for this misunderstanding. We initially included Figure 1B to show that 3’-Seq collected from the nuclear cell fraction represents mRNA that are less subject to decay. This was meant to be a quality control and not speak to whether 3’-Seq is complementary, redundant, nor synergistic with 4sU. In other words, we simply wanted to highlight that nuclear mRNA 3’-Seq and total mRNA 3’-Seq provide different information, in case one might ask whether the two data sets are redundant.

We have now moved Figure 1B to Supplementary file 1 as it simply represents a quality control that we performed.

The description of ribosome occupancy is quite scant. Because intronic polyadenylation would truncate transcripts, change of ribosome occupancy could be simply due to transcript size change (thus ribosome number per nucleotide changes) rather than ribosome number per transcript. The authors need to distinguish these two different scenarios.

While it is true that intronic polyadenylation can truncate transcripts, the apaQTL that we identified as riboQTL were not eQTL. Because ribosome occupancy is calculated in the same way as mRNA expression levels, we would expect apaQTLs resulting in more or less truncated transcripts to be also eQTLs. Furthermore, if transcript length differences accounted for the changes in ribosome occupancy, the shorter isoform would always be estimated to have lower ribosome occupancy and we do not see this pattern.

Reviewer #3:The authors are studying genetic variation that affect APA. While there has been a fair amount of interest in the study of eQTLs the focus on this specific mechanism is less explored. This summer there was a fairly large study of genetic variation and apa published this summer (PMID: 31475030).

We added this as a reference.